# Perceived Exertion: Revisiting the History and Updating the Neurophysiology and the Practical Applications

**DOI:** 10.3390/ijerph192114439

**Published:** 2022-11-04

**Authors:** Thiago Ribeiro Lopes, Hugo Maxwell Pereira, Bruno Moreira Silva

**Affiliations:** 1Laboratory of Exercise Physiology at Olympic Center of Training and Research, Department of Physiology, Federal University of São Paulo, São Paulo 04023-000, SP, Brazil; 2São Paulo Association for Medicine Development, São Paulo 04037-003, SP, Brazil; 3Department of Health and Exercise Science, The University of Oklahoma, Norman, OK 73019, USA

**Keywords:** perception of effort, endurance performance, sports psychology, psychophysiology, training monitoring

## Abstract

The perceived exertion construct creation is a landmark in exercise physiology and sport science. Obtaining perceived exertion is relatively easy, but practitioners often neglect some critical methodological issues in its assessment. Furthermore, the perceived exertion definition, neurophysiological basis, and practical applications have evolved since the perceived exertion construct’s inception. Therefore, we revisit the careful work devoted by Gunnar Borg with psychophysical methods to develop the perceived exertion construct, which resulted in the creation of two scales: the rating of perceived exertion (RPE) and the category-ratio 10 (CR10). We discuss a contemporary definition that considers perceived exertion as a conscious perception of how hard, heavy, and strenuous the exercise is, according to the sense of effort to command the limbs and the feeling of heavy breathing (respiratory effort). Thus, other exercise-evoked sensations would not hinder the reported perceived exertion. We then describe the neurophysiological mechanisms involved in the perceived exertion genesis during exercise, including the influence of the peripheral feedback from the skeletal muscles and the cardiorespiratory system (i.e., afferent feedback) and the influence of efferent copies from the motor command and respiratory drive (i.e., corollary discharges), as well as the interaction between them. We highlight essential details practitioners should consider when using the RPE and CR10 scales, such as the perceived exertion definition, the original scales utilization, and the descriptors anchoring process. Finally, we present how practitioners can use perceived exertion to assess cardiorespiratory fitness, individualize exercise intensity prescription, predict endurance exercise performance, and monitor athletes’ responses to physical training.

## 1. Introduction

The sensations produced during exercise have intrigued scientists from several areas for more than one century [1,2]. Over time, studies investigating exercise-evoked sensations addressed essential topics, including construct validity, measurement properties, neurophysiological mechanisms, and practical applications [2,3,4,5,6]. The studies carried out by the Swedish psychologist Gunnar Borg about the perceived exertion construct from the early sixties onward are a landmark in exercise physiology and sport science [3,4,5,7]. These studies culminated with the creation of two scales extensively used to measure perceived exertion and other exercise-evoked sensations [4]. 

Obtaining perceived exertion is relatively easy, but practitioners often neglect some critical methodological issues extensively addressed when the scales are developed, which can subsequently compromise the validity of perceived exertion assessment [3,4,5]. In addition, several theoretical models have proposed that perceived exertion plays a role in explaining endurance exercise performance [8,9,10,11,12,13,14]. These models rely on assumptions about the origin of the neural signals responsible for generating the perceived exertion [8,11,13,15,16]. Although the scientific knowledge about central and peripheral signals involved in the perceived exertion genesis has notably progressed in the last decade, the scenario is complex, and some caveats remain, requiring an integrative physiological interpretation to advance the field further [11,16]. Lastly, practitioners have extensively applied the perceived exertion to prescribe exercise intensity [17,18,19,20]. However, the practical application of perceived exertion assessment has advanced since its inception [6]. 

For the reasons above, the objectives of the present narrative review are (1) to revisit the history of the perceived exertion construct and scales development; (2) to present available definitions of perceived exertion; (3) to describe potential neurophysiological mechanisms involved in the perceived exertion genesis during exercise, exploring them from an integrative viewpoint; (4) to highlight essential methodological aspects that practitioners should take into account when obtaining perceived exertion; (5) to demonstrate practical applications of perceived exertion assessment during exercise, either in sport or exercise applied to health promotion and rehabilitation programs.

## 2. Revisiting the Perceived Exertion Construct Development

Psychophysics is a psychology discipline that typically investigates the relationship between physical stimuli and sensory responses [3,4,21]. Psychophysics researchers frequently use two experimental approaches: (1) ratio production and (2) magnitude estimation [3,4,7,21,22,23]. In the ratio production method, a subject must produce a physical stimulus proportional (e.g., double or half) to a previously presented reference physical stimulus [4,22]. In the magnitude estimation method, a subject must estimate the magnitude of the sensation generated by a physical stimulus, choosing any number that best represents that sensation [22,23]. For example, a subject can choose the number 10 and another 100 for the same physical stimulus. Using physical stimuli of different magnitudes, it is then possible to establish, with both experimental approaches, the mathematical function that better describes the relationship between physical stimuli and sensory responses [3,4,7,21]. The psychologist Stanley Stevens and his collaborators developed these ratio-scaling methods in the middle of the last century at Harvard University [4,24,25]. Later, other researchers have widely used them, leveraging the research in the psychophysics field [3,4,24,26]. 

Gunnar Borg used the previously presented psychophysics methods to investigate the perceived exertion during exercise [3,4,7,21]. Motivations for this investigation arose from practical observations reported to Gunnar Borg by Hans Dahlström, a Gunnar Borg’s colleague at Umeå University. Hans Dahlström noted that his patients reported a loss of 50% in physical work capacity, but the patients’ performance in a cycle ergometer test had reduced 25% [4]. The initial studies conducted by Gunnar Borg and Hans Dahlström did not focus specifically on the perception of the reduction in physical work capacity over time [3,4,21]. The people did not realize a decline in their physical work capacity but rather an increased effort to perform the same workload [3,4]. Considering that a given workload can generate an overload on skeletal muscles, joints, and the cardiorespiratory system that is proportional to each person’s maximum physical work capacity, Gunnar Borg speculated that the signals coming from the involved sensory receptors would generate a perception of effort proportional to each person’s maximum physical work capacity (i.e., relative exercise intensity). This hypothesis provided an essential theoretical framework for developing scales to measure perceived exertion [4,7,21,27].

Then, in 1959 and 1960, Gunnar Borg and Hans Dahlström investigated the perceived exertion during short-duration (30 s) exercise on a stationary bicycle using the ratio production method [3,5]. In summary, different workloads were applied, and subsequently, subjects were asked to produce half of each of these workloads according to their perceived exertion. Thus, it was possible to establish a power function that mathematically described the relationship between the experimentally imposed workload (physical stimulus) and the perceptually produced workload (sensory response). The exponent of the relationship between imposed workload and perceptually produced workload averaged approximately 1.7. Therefore, this power function would explain the difference between people’s perceived loss and the actual loss in physical work capacity reported by Hans Dahlström to Gunnar Borg. Of note, similar exponent values on the relationship between the workload and perceived exertion were obtained by studies using the ratio production method in different types of exercise (e.g., handgrip) and later, during relatively more prolonged exercise (4–6 min) on a cycle ergometer by using the magnitude estimation method [2,3].

These initial studies helped describe the concept of perceived exertion, but the ratio production and magnitude estimation methods had significant limitations [3,4,7,26]. These techniques did not allow for estimating the absolute level of perceived exertion [3,4,7,26]. For example, a child and a weightlifter can recognize that an object weighs twice as much as another [3,4]. However, this information is irrelevant concerning the absolute effort used to lift the object, which could be different between the child and the weightlifter. Another crucial aspect was the validity of these psychophysical mathematical functions [3,4]. One way to investigate the validity of psychophysical functions would be to test their correspondence with the physiological functions behind the sensory modality in question [3]. However, in the case of perceived exertion assessed without a specific scale, the ensuing psychophysical function showed very low correlations with heart rate [3,26].

In an attempt to overcome these limitations, Gunnar Borg then began to use a category scale with descriptors that anchored the perceived exertion between a minimum and a maximum value [4,7,17,26,27]. He assumed that most people share a similar perception of “maximum effort” (Borg’s range model), even though the absolute physical work capacity achieved in this “maximum effort” was different [3,4,7,17,26,27]. Additionally, evidence at that time showed similar between-people exponents (around 1.6 and 1.7) of the psychophysical relationship between workload and perceived exertion [2,3,17,27]. Consequently, the perceived exertion for a given workload would be proportional to the maximum work capacity of each person [17,27], allowing comparison between individuals (Figure 1). It is also worth noting that Gunnar Borg carefully chose adjectives and adverbs to characterize scale descriptors according to their quantitative semantics properties in such a way that the verbal expressions contained in the descriptors facilitated identifying a level of intensity that converged with the numbers on the scale [7,26].

In the early studies, Gunnar Borg used a 7-point category scale with simple verbal expressions [4]. Later, as some subjects carried out five to seven loads in bicycle ergometer tests, Gunnar Borg increased the numbers on the scale to 21, thus allowing people to have more options to classify the perceived exertion between successive loads [7,28,29]. The 21-point scale, however, exhibited a slightly negative accelerating power function with the workload, which made comparisons with heart rate difficult. Then, a practical observation led to the development of a new scale. Gunnar Borg observed that, on average, a perceived exertion of 17 corresponded to a heart rate of 170 bpm. This coincidence made Gunnar Borg generate a new scale starting on six and ending on 20, corresponding to the resting (60 bpm) and maximum (200 bpm) heart rate of young adults, respectively. The descriptors of the 21-point scale were then mathematically adjusted for the new 15-point scale (from 6 to 20). This new 15-point scale had equidistant intervals so that the effort ratings grew linearly (Figure 2; Panel A), allowing comparison with objective exercise-intensity measurements such as heart rate and oxygen consumption [3,4,21,30]. Thus, a category scale with interval property emerged: the rating of perceived exertion (RPE) scale. 

Following the RPE scale, Gunnar Borg was interested in developing a new scale that would establish the absolute magnitude of sensory response (category rating method) and the mathematical relationship between a physical stimulus and sensory response (ratio scaling method). The development of this new scale had the 7-point category scale as a starting point, as occurred in the creation of the RPE scale. Psychophysics studies at that time showed that category and ratio scales generated different nonlinear growth functions in the sensory responses to physical stimuli of progressive magnitude [4,24,26]. Category scales produced a negatively accelerating growth function, whereas ratio scales exhibited a positively accelerating growth function [24]. Such features allowed Gunnar Borg to mathematically change the verbal descriptors from the 7-point category scale to a ratio scale containing 10 points (Figure 2; Panel B), thus emerging the category-ratio 10 (CR10) scale. The CR10 scale allows reporting decimal numbers (e.g., 0.3) to more finely grade the magnitude of perceived stimuli. In addition, the CR10 scale enables reporting values greater than 10 in case the magnitude of perceived stimuli is higher than the maximal previously experienced, thus avoiding a ceiling effect [7,26]. Other scales were later developed [31,32], such as the category-ratio scale of 100 points [26], but a more detailed description of these scales is beyond the scope of the present narrative review.

## 3. Available Perceived Exertion Definitions

Researchers have recently discussed the meaning of perceived exertion, which may have implications for defining and applying the construct in the practical context [16,33,34]. Gunnar Borg proposed that sensory information from skeletal muscles, joints, the cardiorespiratory system, and any other organ would generate sensations such as pain, fatigue (weakness), strain, and breathlessness. Together, these sensations would form the perceived exertion, a kind of gestalt (i.e., a whole inexplicable by its parts individually) related to the exercise requirement [3,4,7]. Past experiences, expectations about exercise performance, psychological features, environmental conditions, exercise characteristics, and emotions associated with the exercise-evoked sensations would also weigh on the reported perceived exertion [4,35]. Based on these assumptions, Gunnar Borg defined perceived exertion as a “feeling of how heavy, strenuous, and laborious the exercise is” according to the sensation of strain and fatigue in the skeletal muscles and breathlessness or aches in the chest [4]. 

Robert Robertson and Bruce Noble defined perceived exertion as a “subjective intensity of effort, strain, discomfort, and/or fatigue that is experienced during physical exercise” [17]. This definition somewhat agrees with Gunnar Borg’s idea that somatic information from different organs would generate sensations that together would form the perceived exertion. However, people can differentiate several bodily sensations arising during exercise [16,36,37,38,39,40,41,42,43]. For example, if clear instructions are given, it is possible to discriminate the sense of effort to command skeletal muscles from the sensations of force, pain, or discomfort evoked by muscle contractions [16,36,37,39,43]. It is also possible to differentiate the sensations of respiratory effort from the feelings of “air hunger” (insufficient inspiration), breathlessness, and chest tightness [38,40,41]. Such differentiations possibly occur because different neurophysiological mechanisms are involved in the genesis of each of these sensations [2,16,36,37,44]. Thus, considering all exercise-evoked somatic sensations together could hinder the rating accuracy of the perceived exertion [16,30]. 

Samuele Marcora suggested defining perceived exertion as a “conscious sensation of how hard, heavy, and strenuous a physical task is” [8,30,45]. This sensation, however, would depend mainly on the sense of effort to command the involved limbs during the physical task and the feeling of heavy breathing [45]. Given that people can accurately differentiate the sense of effort to command skeletal muscles (locomotor and respiratory) from other exercise-evoked somatic sensations, such as tension, force, pain, discomfort, and breathlessness [16,36,37,38,39,40,41,42,43], it seems that the definition proposed by Samuele Marcora is more accurate for classifying the perceived exertion. It is also worth noting that Marcora’s definition enables quantifying perceived exertion according to the RPE and CR10 scales descriptors, and several studies have shown that perceived exertion is sensitive to different physiological and psychological manipulations using Marcora’s definition [16,30].

## 4. Neurophysiological Mechanisms Associated with Perceived Exertion 

Two theories are frequently used to describe the origin of the neural signals responsible for the genesis of the perceived exertion during exercise [8,15,16,30,45]. One of them, known as the afferent feedback theory (Figure 3), holds that sensory brain areas produce the perceived exertion proportionally to mechanical and metabolic signals detected by receptors in the skeletal muscles and cardiorespiratory system [4,11,13,16]. The same neural signals are also crucial for cardiorespiratory responses to exercise [46,47]. Thus, presumably, perceived exertion and cardiorespiratory responses should be tightly associated. Indeed, several studies have shown high correlations between perceived exertion, heart rate, and pulmonary ventilation responses to exercise [48,49].

Some researchers, however, argue that available evidence does not support the afferent feedback theory [15,16,30]. For instance, beta-blockers or mental fatigue can dissociate perceived exertion and heart rate responses to exercise [15,50,51]. Moreover, information from mechano- and chemo-receptors in the respiratory system (airways, lungs, and chest wall) do not seem to be involved in generating the respiratory effort sensation (i.e., heavy breathing), which is an essential component of perceived exertion during whole-body exercise [15,30,38,52]. Finally, experimental studies that partially blocked group III and IV muscle afferents typically do not show changes in perceived exertion during exercise compared to a control condition [15,30,53,54,55]. Nevertheless, a cautious interpretation is required, considering that some of the mentioned studies were not specifically designed to investigate the neurophysiological mechanisms behind the perceived exertion during exercise [53,54,55]. 

In contrast to the afferent feedback theory, the corollary discharges theory (Figure 3) proposes that the signals that generate the perceived exertion come from efferent copies associated with the motor command to locomotor muscles and the central drive to respiratory muscle [15,30,33]. Specifically, outputs from the supplementary motor area and medullary respiratory center are sent directly to sensory areas. These signals are parallel (i.e., corollary discharges) and somewhat independent of the signals sent to locomotor and respiratory muscles [8,15,16,30,33,45,56,57,58,59,60]. In support, experimental findings have shown that perceived exertion accompanies the changes in motor-related cortical potential (a proxy to central motor command) induced by manipulations that do not alter afferent signals (e.g., use of caffeine or eccentric exercise-induced force reduction) [56,57]. Therefore, these findings indicate that corollary discharges, rather than afferent signals, are vital to the generation and modulation of perceived exertion. However, researchers opposed to the afferent feedback theory have not considered that the redundancy and interaction between neurophysiological mechanisms are crucial for cardiovascular and respiratory adjustments to exercise [46,47,61]. Such a phenomenon likely contributes to the formation of perceived exertion as well. Thus, next, we propose how both theories might physiologically operate together, which should be taken into account by future studies.

Recent evidence has suggested that the genesis of perceived exertion during high-intensity exercise can indirectly involve the afferent feedback from the skeletal muscle and cardiorespiratory system. For example, the activation of group III and IV afferents receptors in the locomotor muscles by metabolite accumulation reduces the excitability of the primary motor cortex, hindering muscle recruitment [62,63,64,65,66]. In this case, the supplementary motor area has to increase the signals to the primary motor cortex to preserve the muscle power output, which provides additional corollary discharges for the genesis of the perceived exertion [16,56,57,58,60,67]. In addition, the elevated respiratory work may generate metabolite accumulation in the respiratory muscles, which also activates underlying group III and IV afferent fibers [68,69,70]. The activation of respiratory muscle afferents leads to sympathetically-mediated vasoconstriction that impairs the oxygen supply to the locomotor muscles, exacerbating metabolite accumulation in both locomotor and respiratory and ultimately inducing primary motor cortex inhibition [62,63,64,65,66,68,69,70]. Again, enhanced activation of the supplementary motor area would be required to sustain the muscle power output, potentially increasing corollary discharges. Supporting evidence is the increase in diaphragmatic muscle activation (i.e., EMG response) associated with a decline in evoked transdiaphragmatic twitch pressure during an incremental exercise [71]. In this scenario, an increase in medullary corollary discharges would also contribute to the rise in the perceived exertion.

## 5. Methodological Issues to Quantify Perceived Exertion

Some methodological issues are fundamental to quantifying the perceived exertion during exercise accurately. One of these issues is using the original versions of the scales, regardless of whether it is Borg’s scale or not. The previous section showed that the psychophysics properties of the RPE and CR10 scales were carefully verified and validated over several years. It is, therefore, inappropriate altering the RPE and CR10 scales by using figures, colors, other non-tested verbal descriptors, or verbal descriptors for all scale numbers [16,26]. Considering that the scales were developed in English, if using the scales in another language, it is also strongly recommended to verify if the translated version has passed a thorough transcultural validation process [72]. Translated versions of the RPE and CR10 in various languages are available from the Swedish company website licensed to distribute Borg’s scales (https://borgperception.se, accessed on 1 November 2022). Moreover, practitioners should ideally obtain perceived exertion during exercise [4,16]. If not possible, an option is obtaining values immediately after exercise. However, practitioners should remind the tested individual to report values referring to the exercise performed [16].

Providing written instructions when obtaining perceived exertion was a methodological procedure originally recommended by Gunner Borg [4], which a review article recently reinforced [16]. Given that these instructions were developed in English, transcultural adaptation to other languages should also be considered [72]. Similar to Borg’s scales, translated versions of the instructions are also available (https://borgperception.se, accessed on 1 November 2022). However, practitioners should be aware that the instructions were developed using Borg’s definition of perceived exertion [4,72]. As we pointed out in the previous paragraphs, contemporary studies support that it is essential to distinguish effort sensation to command skeletal muscles (locomotor and respiratory) from other exercise-evoked somatic sensations (e.g., pain or breathlessness). Clear instructions differentiating the sensations can be critical for accurate perceived exertion quantification [16].

In the written instructions, individuals should be oriented first to read the descriptors and then to quantify the perceived exertion [4,16]. In the case of the CR10 scale, individuals should be encouraged to report decimal values (e.g., 0.5), thus grading more finely the perceived exertion magnitude [4,16,26,30]. Providing an example of maximal perceived exertion to the individuals is strongly recommended. This anchoring process can be done based on the individual’s memory or the individual’s experience with a performed exercise [4,16,51]. In the case of the CR10 scale, values above 10 are possible if the current perception is more intense than previous experiences. The anchoring process and the clear construct definition are essential procedures for a valid measure of perceived exertion [17]. Lastly, it is worth reminding the individuals to be as honest as possible and avoid comparing with others. It is also encouraged to avoid judgments about exercise intensity that can result in the tested individual underestimating or overestimating the reported perceived exertion [4].

## 6. Practical Applications of Perceived Exertion Measurement

Maximal oxygen uptake and peak exercise intensity are frequently used to assess cardiorespiratory fitness and individualize exercise prescription, respectively [73,74]. It is possible to estimate these parameters through the perceived exertion when it is undesirable to push the incremental exercise testing until the subject’s voluntary exhaustion [6,75,76,77]. It is only necessary to extrapolate the submaximal relationship between perceived exertion and oxygen uptake or exercise intensity to a theoretical endpoint (i.e., 19 or 20) on the RPE scale (Figure 4). It is also possible to estimate the time to exhaustion during constant-load exercise tests, given the linear relationship between perceived exertion and exercise time [9,78]. Additionally, the product between perceived exertion and remaining distance fraction (i.e., hazard score) can predict the subsequent running speed change during time-trials tests (e.g., completing 5 km in the shortest time). Hazard scores below 1.5 and above 3 arbitrary units are associated with a reduction and an increase in the running speed, respectively [79].

Critical power delimits the transition between heavy and severe exercise intensity domains [80,81]. In the exercise above the critical power (i.e., severe-intensity domain), fatigue-related metabolites accumulate (e.g., inorganic phosphate and hydrogen ions) over time in the skeletal muscle [81,82,83], limiting the capacity to sustain the exercise for a prolonged time [81]. Therefore, critical power is a valuable tool for continuous or interval endurance training prescription, and it is considered an important indicator of performance in endurance sports [80,81,84]. It is possible to approximate the critical power through perceived exertion slopes obtained from three or more constant-load exercise tests in the severe-intensity domain [85,86,87]. The intercept between exercise intensity and perceived exertion slopes approximates the critical power (Figure 5). Importantly, this method of critical power approximation does not require exercise until exhaustion, as the perceived exertion slopes can be obtained from intermediate levels (11–14 on the RPE scale) of perceived exertion [87].

Monitoring athletes’ responses to training (i.e., training effect) can provide valuable information to refine the training process, maximizing the chances of improving sports performance and minimizing the risk of injury, illness, nonfunctional overreaching, or overtraining [88,89]. It is possible to track physical fitness changes by quantifying the workload during incremental exercise testing corresponding to a specific level of perceived exertion (e.g., 15 or 17 on the RPE scale) [17,50]. For example, an increased workload to the same level of perceived exertion represents an improvement in physical fitness—i.e., a positive training effect [17,50]. In addition, when athletes are in a state of accumulated fatigue due to an imbalance between training loads and recovery periods, a lower heart rate accompanies a higher perceived exertion for a given workload [90,91]. Heart rate reduction for the same exercise intensity most often indicates positive training adaptations [92,93]. Therefore, measuring perceived exertion simultaneously with heart rate during constant-load exercise tests seems to permit more accurate monitoring of the athletes’ responses to training.

The aforementioned practical applications are based on measuring perceived exertion during an externally imposed-exercise intensity (estimation approach). An alternative approach would be self-regulating exercise intensity while maintaining a given perceived exertion over time (production approach). For example, evidence suggests that incremental exercise testing self-regulated by perceived exertion can produce similar values of maximal oxygen uptake and ventilatory threshold compared with traditional protocols, but it seems unfeasible to determine the respiratory compensation point [94,95,96]. Moreover, several studies have also shown that self-regulation of exercise intensity by perceived exertion produces similar cardiorespiratory and metabolic responses to those obtained during incremental exercise testing for the same perceived exertion [17,19,20,97]. It is, therefore, possible to use the perceived exertion corresponding to percentages of maximal oxygen uptake or maximal heart rate obtained in incremental testing (e.g., 60% and 80%) to control exercise intensity during training sessions [17,20], which has important practical implications when prescribing exercise for health or throughout rehabilitation programs.

## 7. Conclusions

Early studies conducted by Gunnar Borg about the perceived exertion construct used psychophysics methods (ratio production and magnitude estimation). Given the limitations of these methods (impossibility of between-individuals comparison and low correlations with physiological variables), Gunnar Borg developed a scale with categorical descriptors anchored within a minimum and a maximum limit for perceived exertion judgment (Borg’s range model) in subsequent studies. These later studies gave rise to the RPE and CR10 scales. Perceived exertion should be defined as a conscious perception of how hard, heavy, and strenuous the exercise is, emphasizing that perceived exertion depends only on the sense of effort to command the limbs and the feeling of heavy breathing (respiratory effort). This contemporary definition is related to neurophysiological mechanisms involved in the perceived exertion genesis. Regarding neurophysiological mechanisms, efferent copies from the motor command and respiratory drive (i.e., corollary discharge) appear directly linked with the generation of perceived exertion. On the other hand, feedback from group III and IV muscle (locomotor and respiratory) afferents might indirectly participate in the perceived exertion genesis during high-intensity exercise, modulating the magnitude of corollary discharges. 

Some methodological issues are fundamental to quantifying the perceived exertion accurately. One of these issues is using an updated definition of perceived exertion proposed by Samule Marcora. Thus, other exercise-evoked sensations would not hinder the accuracy of perceived exertion assessment. We strongly recommend using the original scales versions, regardless of whether it is Borg’s scales or not, since the psychophysics properties of the RPE and CR10 scales (and others) were carefully verified and validated over several research years. Another essential issue is the evaluator providing the assessed person with an example of maximal perceived exertion. This anchoring process can be done based on the individual’s memory or individual’s experience with a performed exercise. When carefully applied, exercise and sports science practitioners can use the perceived exertion during incremental, constant-load, and time-trial exercise testing to assess cardiorespiratory fitness, prescribe individualized exercise intensities, predict endurance exercise performance, and monitor athletes’ responses to physical training. Therefore, determining perceived exertion during exercise has important implications for health promotion, rehabilitation programs, and high-performance sports. 

## Figures and Tables

**Figure 1 ijerph-19-14439-f001:**
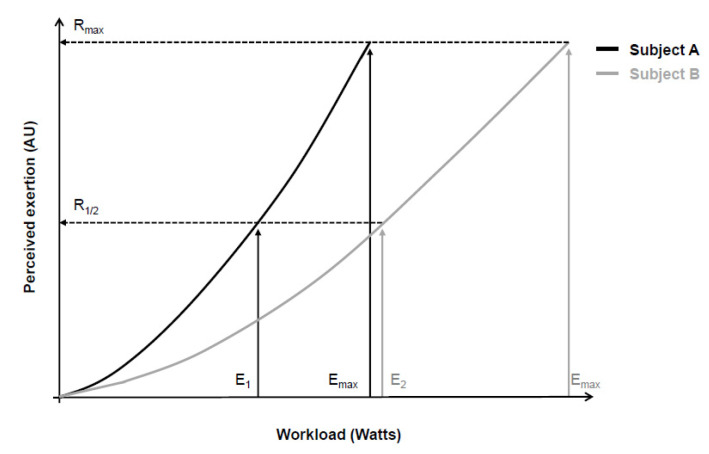
Psychophysical relationship between workload during a cycling exercise (physical stimulus) and perceived exertion (perceptual response) of two subjects (continuous black and gray lines). According to the range model proposed by Gunnar Borg, subjects’ maximum perceptual response (R_max_) should be equal, despite between-subjects differences in maximal physical stimulus (E_max_). Consequently, different absolute physical stimuli (E_1_ and E_2_) correspond to similar relative perceptual responses (R_1/2_). Adapted from Marks, Borg, and Ljunggren [27].

**Figure 2 ijerph-19-14439-f002:**
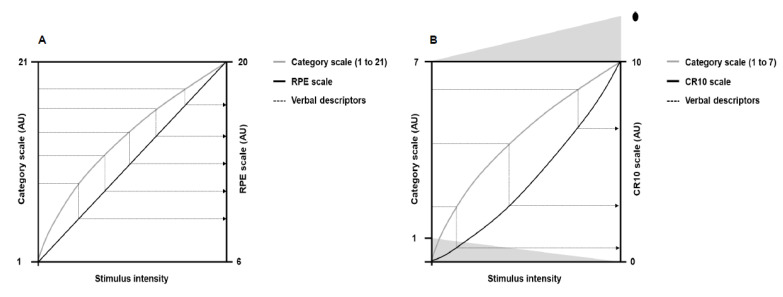
Illustration of the theoretical principle behind the verbal descriptors shifts (dotted lines) from the 21-points and 7-point category scales to the rating of perceived exertion (RPE) and category-ratio 10 (CR10) scales, respectively. Panel (**A**) shows perceived exertion responses to the 21-point scale (gray line) and the RPE scale (black line) as a function of the physical stimulus intensity. Panel (**B**) shows perceived exertion responses to the 7-point category scale (gray line) and the CR10 scale (black line) as a function of the physical stimulus intensity. The dot above number 10 represents a number people can choose to quantify maximal exertion. Adapted from Borg [4].

**Figure 3 ijerph-19-14439-f003:**
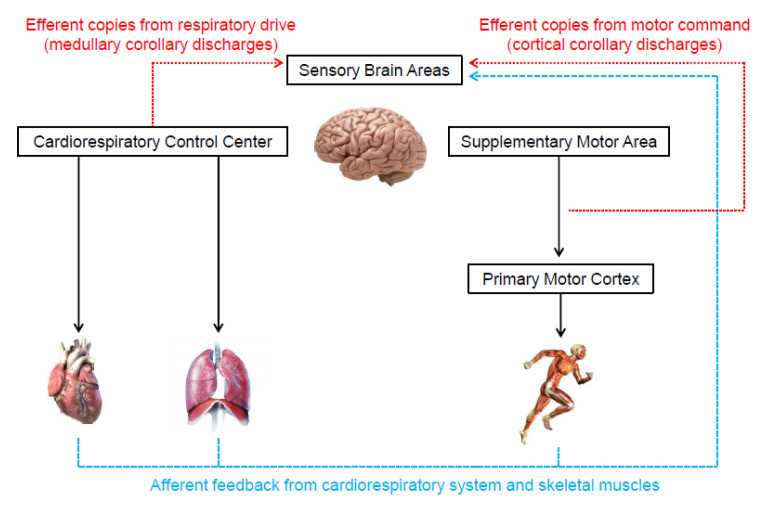
Illustration of the neurophysiological mechanisms underlying the genesis of perceived exertion during exercise.

**Figure 4 ijerph-19-14439-f004:**
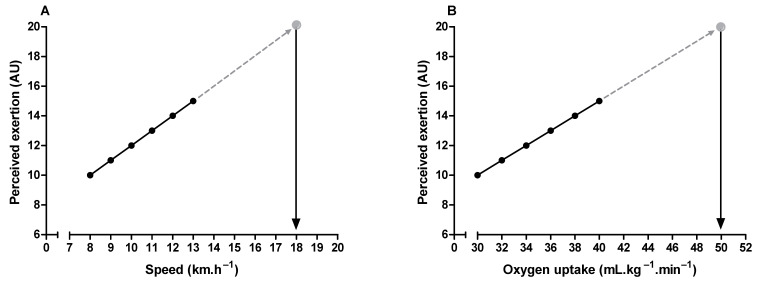
Illustration of peak exercise intensity (**A**) and maximum oxygen uptake (**B**) estimation through perceived exertion measurement during submaximal incremental exercise testing.

**Figure 5 ijerph-19-14439-f005:**
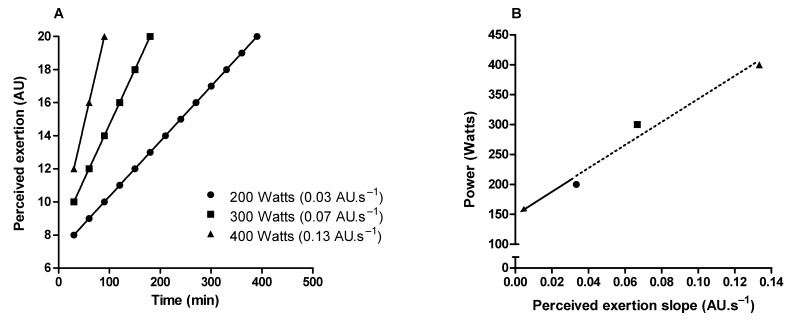
Illustration of critical power determination by obtaining the perception of exertion throughout constant load tests in the severe-intensity domain. (**A**) Shows perceived exertion over time in three constant-load exercise tests. (**B**) Shows the linear relationship between the power used in the constant-load exercises tests and the rate of perceived exertion increase over time (perceived exertion slope). Critical power is the intercept (arrow) of the linear regression.

## Data Availability

Not applicable.

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
