# Peer review of "Perceived Exertion: Revisiting the History and Updating the Neurophysiology and the Practical Applications"

_ijerph, 2022, doi:10.3390/ijerph192114439_

Round 1
Reviewer 1 Report
Thank you for the opportunity to read this well-written paper. I wish to congratulate the authors for their work. This paper is a significant contribution to the field, and I enjoyed reading it.
Only one remark:
Line 81: maybe “only” is not good word, 25% reduction is significant; consider omitting or using another word. But this is just a suggestion
Reviewer 2 Report
Review of the manuscript entitled: Perceived Exertion: Revisiting the History and Updating the Neurophysiology and the Practical Applications The manuscript submitted is appropriate to the subject matter and scientific rigor. The authors raised a very current issue at work, which is not only interesting from a scientific but also a practical point of view. Some remarks improving the quality of future research. and suggested changes and comments to the submitted manuscript in order to improve the quality of the planned research and future publications below:
1. Would you please correct the bibliography in accordance with the journal's guidelines and standardize it. Please provide electronic access to all position in references. In bibligrafi, add Issue and numbers DOI to all items.
2. Would you please cheek and correct references: [27], [39], [41], [74], [75], [91], [93] and [95]
3. Enter references (items in the bibliography) for:
a) The psychologist Stanley Stevens developed these ratio-scaling 73 methods in the middle of the last century at Harvard University ... ..line number 73
b) ... by Hans Dahlström ...line number 78
c) The initial studies conducted by Gunnar Borg and Hans Dahlström ...line number 81
Remaining comments in the article in the attachment.

Reviewer 3 Report
First, congratulate the authors for the work done and for the author's chosen theme.
It is a critical review of the relevant literature with well-defined objectives.
However, the critical review of the literature should not be limited to presenting the existing literature, and it should add something new starting from the reflection on it. It is precisely this question that, from my point of view, weakens the work reviewed here. We finished reading and thought, "What is the difference between what I am reading and what I already know?". The literature review is not a mere synthesis of what is already found in previous research. Authors need to make clear what they add to the literature. Where is the novelty in their reflection?
For example, in the objective of their work, the authors put "to present the contemporary definition of perceived exertion," and I expected a definition proposal from the authors themselves. However, they only present more recent definitions by other authors, one of the definitions dating back to 1997.
Round 2
Reviewer 3 Report
I read all the authors' answers carefully, and I understand the study's relevance. However, from my point of view, it is incorrect to state that objective 2 of this study is "2) to present a contemporary definition of perceived exertion". The paper did not introduce a new vision or definition of perceived exertion. The authors only recover and work on the vision of Samuele Marcora dating from 2009. The authors themselves state this in their reply letter. the paper revisit a definition rather than display a definition. Those who read the paper look for a new definition due to the promise of the goal, and this is different. It is necessary to change objective two, or this goal will error the reader.
